# Body Composition and Bioelectrical-Impedance-Analysis-Derived Raw Variables in Pole Dancers

**DOI:** 10.3390/ijerph182312638

**Published:** 2021-11-30

**Authors:** Giada Ballarin, Luca Scalfi, Fabiana Monfrecola, Paola Alicante, Alessandro Bianco, Maurizio Marra, Anna Maria Sacco

**Affiliations:** 1Department of Movement Sciences and Wellbeing, University of Naples “Parthenope”, 80133 Naples, Italy; giada.ballarin001@studenti.uniparthenope.it; 2Department of Public Health, School of Medicine, Federico II University, 80131 Naples, Italy; scalfi@unina.it (L.S.); fabiana.monfrecola@hotmail.it (F.M.); paola.alicante@unina.it (P.A.); alessandrobianco92@gmail.com (A.B.); 3Department of Clinical Medicine and Surgery, School of Medicine, Federico II University, 80131 Naples, Italy; marra@unina.it

**Keywords:** bioelectrical impedance analysis, muscle composition, phase angle, impedance ratio, pole dance

## Abstract

Few data are available on the body composition of pole dancers. Bioelectrical impedance analysis (BIA) is a method that is used to estimate fat-free mass (FFM) and fat mass (FM), while raw BIA variables, such as the impedance ratio (IR) and phase angle (PhA), are markers of body cell mass and the ratio between extracellular and total body water. The aim of this study was to evaluate the body composition of pole dancers compared to controls, in particular, those raw BIA variables that are considered as markers of muscle composition. Forty female pole dancers and 59 controls participated in the study. BIA was performed on the whole body and upper and lower limbs, separately, at 5, 50, 100 and 250 kHz. The FFM, FFM index, FM and body fat percentage (BF%) were predicted. The bioelectrical impedance indexes IR and PhA were also considered. Pole dancers exhibited higher FFMI and BI indexes and lower BF%. PhA was greater and IRs were smaller in pole dancers than in controls for the whole body and upper limbs. Considering the training level, FFM, whole-body IR and PhA were higher in the professionals than non-professionals. Raw BIA variables significantly differed between the pole dancers and controls, suggesting a higher BCM; furthermore, practicing pole dancing was associated with a greater FFM and lower FM.

## 1. Introduction

The evaluation of body composition is crucial not only for assessing nutritional status in the general population but also for athletes for the monitoring of training and performance.

Anthropometry and bioelectrical impedance analysis (BIA) are both field methods that are widely used to assess the body composition of athletes [1]. In particular, BIA is a simple, non-invasive technique that measures the electrical characteristics of the human body, i.e., impedance (Z) and phase angle (PhA) (from those, resistance (R) and reactance (Xc) can also be derived). Total body water (TBW), fat-free mass (FFM) and fat mass (FM) can be estimated by means of predictive equations that include BIA variables and very often other variables, such as age, height and body mass; some equations were specifically developed for athletes. Since these specific equations [2,3,4] have not been definitively validated, the BIA-derived estimation of body composition should be considered with caution. In particular, the BIA method has an error of 4–8% compared to criterion methods, which could be even more evident in athletes [3]. On the other hand, the BIA estimates of body composition might give some interesting evidence on body composition on a groupwise basis. Of note, the bioimpedance index at 50 kHz (BI index = height^2^/Z at 50 kHz) is commonly considered as a logical predictor of FFM and TBW [5,6]. Finally, it should be noted that BIA may be performed on the whole body but also separately for upper limbs and lower limbs (segmental BIA), giving, at least in theory, the chance for evaluating appendicular muscle mass [7,8,9].

Raw BIA variables, such as the impedance ratio (IR), which is the ratio between Z at high frequencies and Z at low frequencies, and PhA at 50 kHz, are those that are directly measured by an analyzer. Their assessment in sportspeople is motivated by the fact that IR and PhA may be considered as potential markers of both body cell mass (BCM) and the ratio between extracellular water and total body water (ECW/TBW ratio) [10,11,12,13]; in other words, both these variables may give information on the electrical properties, as well as the FFM composition and/or muscle composition. IR and PhA were related to muscle strength and physical activity [14,15] as well, while in the first decades of life and elderly people PhA was associated with muscle performance [16,17], isolated or grouped physical fitness indicators [18,19], and cardiorespiratory fitness [20]. As reported in a recent systematic review [21], it is still to be determined to what extent PhA differs between different sports and due to training/untraining; some studies showed that mean whole-body PhA is higher in athletes vs. controls [21,22], while, to the best of our knowledge, so far no data are available on IRs in sportspeople and only limited data exists on segmental BIA [7,8,9,21,22].

With regard to sports activities, pole dancing is a type of functional training that involves the use of a vertical pole to perform exercises and figures. A training session, called a pole class, lasts between 60 and 90 min (possibly depending on training level) and can be subdivided into three parts: warm-up and strengthening exercises are performed first; then the specific tool figures are studied, with increasing difficulty of execution, while cooldown exercises close the session. Pole dancing may be considered a moderate-intensity cardiorespiratory endurance exercise that, if practiced regularly, leads to a significant increase in aerobic capacity, resistance, flexibility, and motor coordination [23,24].

To the best of our knowledge, only a single study has evaluated the body composition of female pole dancers, attributing an increase in postural strength and stability to the more experienced athletes, but no changes in body composition [25]. Looking at similar sports, rhythmic gymnasts exhibited lower body mass, body mass index (BMI) and skinfold thickness compared to other athletes [26], while gymnasts had a reduced body fat percentage (BF%) compared to controls with the same BMI [27,28]. Dancers had similar BF% but higher levels of FFM and muscle mass than controls, whereas low values of FFM and fat mass (FM) were observed in the case of underweight athletes [29]. Finally, in sedentary women, a choreographed fitness group workout contributed to reducing FM and increasing muscle mass [30].

Against this background, the aim of this cross-sectional study was to evaluate the body composition of pole dancers (non-professional and professional athletes) compared to controls, with a particular interest in the raw BIA variables that are thought to be markers of FFM composition and/or muscle composition. In addition, a segmental BIA evaluation was performed to explore the electrical characteristics of upper or lower limbs.

## 2. Methods

Forty female pole dancers and fifty-nine control young women participated in the study. Pole dancers were recruited from among those going to two gyms in Naples (a participation rate of 89%) and were non-professional performers (hereafter defined as amateurs) (*n* = 33), who trained 2–4 h a week in two sessions (18–36 months of specific training), and professionals (*n* = 7) who were pole dance trainers (at least 60 months and more than 6 h a week of specific training). Controls (*n* = 59) were sedentary women (at most 1 h of physical training twice a week) and were recruited from among the female students attending the “Federico II” University of Naples. All subjects were healthy. The Ethics Committee of the “Federico II” University of Naples approved the research protocol and subjects gave their informed consent to participate in the study.

The participants avoided physical exercise for 24 h before the measurement session and were studied by the same operator following standard procedures. Data were collected between March and April 2019 in four sessions for pole dancers and six sessions for controls (data on ≥10 women were collected in each session). The general schedule was similar in the two groups of pole dancers, with different intensities of training programs based on their training level.

Body mass was measured to the nearest 0.1 kg using a platform beam scale and height was measured to the nearest 0.5 cm using a stadiometer (Seca, Hamburg, Germany). Participants were asked to remove shoes and heavy clothes prior to weighing. BMI was then calculated as body mass (kg)/height^2^ (m^2^).

Height was measured according to standard procedures. The participants were asked to stand up straight against the backboard with their body weight evenly distributed and both feet flat on the stadiometer platform, while the head was in the Frankfort horizontal plane [31].

Mid-arm circumference and triceps skinfold thickness (Holtain skinfold caliper) were measured on both body sides and, subsequently, the arm muscle area (AMA), corrected for the bone area, and arm fat area (AFA) were calculated as follows [32]:AMA = [(Mid-arm Circumference − π × TSF) × 2/4π] − 6.5
AFA = Arm total area − AMA

BIA was performed using a HUMAN IM TOUCH multi-frequency analyzer (DS MEDICA, Milan, Italy) in standardized conditions: ambient temperature between 23–25 °C, fast for >3 h, empty bladder and supine position for 10 min. Data on Z at four different frequencies (5, 50, 100 and 250 kHz) and PhA at 50 kHz were considered for the statistical analysis. Precision resistors and capacitors (reference electronic circuits) were routinely used for calibration. The reproducibility of the BIA was previously assessed in ten healthy volunteers on subsequent days with a mean coefficient of variation of 1.5% for Z (at each of the different frequencies considered) and 2% for the phase angle at 50 kHz.

The 250 kHz/5 kHz IR may be used as a proxy marker of fluid distribution and was recently related by our group to mortality in patients with chronic obstructive pulmonary disease [10,14]. Subjects were asked to lie down with their legs and arms slightly abducted (~30°) to ensure no contact between body segments. The measuring electrodes were placed on the anterior surface of the wrist and ankle, and the injecting electrodes were placed on the dorsal surface of the hand and the foot, respectively [13]. Segmental BIA was performed using a six-electrode technique according to Organ [33].

Whole-body BI indexes were calculated as height^2^ divided by Z as markers of ECW (Z at a low frequency of 5 kHz) and FFM (Z at high frequencies of 50, 100 or 250 kHz). In addition, two other raw variables were measured for the whole body and upper or lower limbs separately: (1) IR is commonly calculated as the ratio between Z at 200, 250 or 300 kHz and Z at 5 kHz [10]. In the present study, data were obtained for three ratios: Z 50 kHz/Z 5 kHz (IR50/5), Z 100 kHz/Z 5 kHz (IR100/5), and Z 250 kHz/Z 5 kHz (IR250/5). (2) PhA was measured at 50 kHz, as described in the literature. To the best of our knowledge, there has been little interest in applied physiology and human nutrition for evaluating the phase angle at frequencies other than 50 kHz. In all cases, mean values for the dominant (D) and non-dominant (ND) body sides were considered for statistical analysis to give more consistent results for the entire body. FFM was estimated using the Sun equation [34], which is a well-known equation that was proposed for the general population aged 12–94 years and which is also expected to perform well in young women with a higher physical activity level but no very major changes in body composition.

Whole-body FFM was calculated as follows:FFM = −9.53 + 0.69 × height^2^/resistance + 0.17 × body mass + 0.02 × resistance
where the resistance at 50 kHz was derived by multiplying Z by the cosine of PhA.

Finally, FM was obtained from the difference between body mass and FFM, while the fat-free mass index (FFMI) was calculated as FFM (kg)/height^2^ (m^2^).

### Statistical Analysis

Data obtained during the routine examination of athletes or control subjects were retrospectively retrieved. With a type I error rate of 0.05 and a type II error rate of 0.20, a sample size of 85 subjects is required to determine whether a correlation coefficient of 0.3 differs from zero.

Results are expressed as mean ± standard deviation (with some exceptions, see below). Statistical significance was pre-determined as *p* < 0.05. Effect size was calculated according to Cohen [35].

All statistical analyses were carried out using the Statistical Package for Social Sciences (SPSS Inc., Chicago, IL, USA) version 26. One-way analysis of variance was performed to assess the differences between two groups (pole dancers vs. controls or amateurs vs. professionals). Partial correlation was used to assess the relationships between the variables. The general linear model (GLM) was used to assess how several variables affected the continuous variables. From a practical point of view, it was used to compare the body composition between groups after controlling for body mass; adjusted means ± standard errors were provided by this statistical procedure.

## 3. Results

The general characteristics of the study groups are summarized in Table 1. Despite no difference in body mass and BMI, the pole dancers exhibited lower BF% compared to the controls (−14%). Correspondingly, the AMA was significantly greater and the AFA was smaller in the pole dance than in the control group (Table 1).

As for the raw BIA variables, the whole-body and upper limb Z values were lower in the pole dancers than in the controls; for instance, Z at 250 kHz was 485 ± 50 vs. 519 ± 38 kHz and 240 ± 28 vs. 271 ± 20 kHz, respectively (d = 0.39 and d = 0.72; *p* < 0.001), with small differences (<2%) between the D and ND body side. Furthermore, Table 2 indicates that the BI indexes at 5, 50, 100 and 250 kHz were higher in the pole dancers than in the controls (+4.3, +4.9, +5.3 and +5.3%, respectively). These differences in the mean values of different Z and BI indexes persisted after adjusting for age and mass (data not shown). After controlling for groups, a partial correlation indicated that whole-body BI indexes were associated with AMA (r > 0.450 for 50, 100 and 250 kHz vs. r = 0.416 for 5 kHz) but not with AFA.

As shown in Table 2, PhA was greater in pole dancers than in controls by 3.8% for the whole body (d = 0.39 and *p* = 0.063) and by 10.7% for upper limbs (d = 0.89 and *p* < 0.001), whereas there was no difference for lower limbs. IRs were lower in the pole dance group than in the control group, again more markedly for upper limbs (Table 2). The differences for upper limbs were still found in both cases even after controlling for age and body mass. In particular, multiple regression analysis indicated age and body mass as predictors of IR250/5 (for the whole model: R^2^ = 0.117, F(2,87) = 6.83, *p* = 0.002) and PhA (R^2^ = 0.053, F(2,87) = 5.90, *p* = 0.017). Of note, no relationships were detected between IRs or PhA and body composition.

There was no significant association of PhA or IRs with height, mass, BMI, FFM, FM, AMA or BI indexes (*p* > 0.20, data not shown). On the other hand, after adjusting for groups, a partial correlation indicated a moderate association between the upper limb and lower limb values of PhA (r = 0.463), IR50/5 (r = 0.538), IR100/5 (r = 0.531) and IR250/5 (r = 0.514).

With respect to the training level, professional and amateur pole dancers did not differ in terms of body mass (55.6 ± 4.2 vs. 57.3 ± 7.3 kg) and BMI (22.0 ± 2.3 vs. 22.2 ± 2.3 kg/m^2^). The GLM indicated that, after adjusting for body mass, FFM (mean ± SEM, 45.3 ± 0.6 vs. 43.7 ± 0.3 kg, *p* = 0.024) was greater in the more trained than in the less trained athletes, while BF% was smaller (21.4 ± 11.1 vs. 24.2 ± 0.5%, *p* = 0.023, respectively). In particular, multiple regression analysis was used to test whether training level and body mass significantly predicted participants’ FFM and BF%. The results indicated that the two predictors explained 75% of the total variance for FFM (R^2^ = 0.75, F(2,86) = 130.9, *p* < 0.001) and 64% of the total variance for BF% (R^2^ = 0.64, F(2,86) = 78.0, *p* < 0.001).

Turning to raw BIA variables, whole-body PhA and IRs were higher, but not significantly (d between 0.5 and 0.8; *p* between 0.05 and 0.10), in the professional athletes than in the amateur athletes (Table 3). More evident differences (Figure 1) emerged for the upper limbs: the professional pole dancers had significantly smaller IRs and greater PhA than the amateur athletes and controls, and the same was true when amateurs were compared to the controls (d = 0.99 and *p* < 0.05). After taking into consideration the training level as a predictor, no significant relationships were found between IRs or PhA and body mass or body composition.

## 4. Discussion

In the present study, raw BIA variables that may be considered as markers of FFM composition and/or muscle composition significantly varied between female pole dancers and controls, showing different electrical characteristics of the body and suggesting higher BCM; in addition, pole dancers exhibited lower BIA-derived FM and BF%.

We performed a cross-sectional study on a relatively large group of pole dancers compared to sedentary controls, bearing in mind that the effects of this type of training on body composition have so far been poorly explored [25]. Unfortunately, there was no information regarding participants’ body composition before starting the training. Indeed, in light of the difficulties in carrying out long-term intervention studies, the present cross-sectional study is expected to provide some preliminary insights regarding the effect of pole dancing on body composition.

Body composition was assessed using BIA, which is a technique that is widely used in athletes [1]. Since the specific equations developed for athletes [2,3,4] have not been definitively validated [3,13], BIA-derived estimation of body composition should be considered with caution. In particular, the BIA method has an error of 4–8% compared to criterion methods, which could be even more evident in athletes [3]. On the other hand, the BIA estimates of body composition might give some evidence on body composition on a groupwise basis. In the present study, the Sun equation was chosen to predict FFM [34]; this formula was developed in a large sample of healthy subjects using a multicomponent model, it is widely used, and it is expected to also perform well in young women with a higher physical activity level but no major changes in body composition.

Thus, we looked first at BIA-derived estimates of body compartments. Despite having similar body mass and BMI, pole dancers had lower FM and BF% compared to the controls. These findings are in agreement with those reported in previous cross-sectional studies that showed higher FFM and smaller FM in female gymnasts and dancers [26,27,28]. Of note, the study by Nawrocka et al. [25] on the body composition of pole dancers did not include a control group. Overall, our results suggest a significant, but small effect of pole dance training on body composition, with a moderate to high effect size for BF% (d = 0.74 and *p* = 0.001).

As an alternative approach, IRs and PhA (for the whole body and upper and lower limbs, separately) were directly (no predictive equations used) determined in pole dancers and controls as a qualitative approach to body composition analysis [13]. Both those raw BIA variables may be effective in exploring FFM composition and muscle composition in terms of the electrical characteristics of tissues, as well as BCM and the ECW/TBW ratio [10,11,12,13]. Interestingly, IRs and PhA have also been associated with muscle strength and physical activity [14,15,19]. A few cross-sectional studies showed that mean whole-body PhA is higher in athletes vs. controls, while, to the best of our knowledge, no data so far are available on IRs [21]; of note, a recent paper showed, as expected, a high correlation between IRs and PhA [19]. In addition, it is still to be determined to what extent IR and/or PhA may vary between different sports and with training/untraining [13,21]. Facing this background, although in our experience data on IR or PhA are very reproducible, the use of these BIA variables in longitudinal studies or single athletes should be better defined and considered with caution.

IR is commonly calculated as the ratio between Z at high frequency and Z at low frequency [10]. The ratio between Z at 200 kHz and Z at 5 kHz (IR200/5) is widely used but still not formally indicated as the only one to be taken into consideration. Results on three different IRs are reported here, with IR250/5 being very close to IR 200/5. The three IRs were all slightly smaller in the pole dance group compared to the control group. At first glance, these differences in IRs were small in percentage terms, but relevant when compared to the corresponding standard deviations. For instance, the difference in IR250/5 was 0.007, while the pooled standard deviation was 0.019 (d = 0.39 and *p* = 0.058). Regarding another raw BIA variable, whole-body PhA, which was measured at 50 kHz, as commonly described in the literature [10,21], was only slightly higher in pole dancers compared to controls (low size effect). Overall, only minor changes were observed for the whole body.

It is clear that segmental BIA, as performed on upper limbs and lower limbs separately, may give, at least in theory, the chance to evaluate appendicular muscle mass more directly [7,8,9]. Few previous papers have performed this type of measurement in athletes; they, for instance, showed greater PhA for both lower and upper limbs in female volleyball players compared to controls [8]. Our study yielded some results of interest: lower limb IRs and PhA did not differ between the groups, while a marked difference emerged for upper limbs (d = 0.72 and *p* < 0.001 and d = 0.89 and *p* < 0.001, respectively), suggesting some effects of pole dancing on different muscle groups. Of note, those differences persisted after adjusting for age plus body mass or plus body composition. Thus, segmental measurement seemed to be effective in detecting differences in raw BIA variables, which should be examined in detail by further studies that consider various types of training and use different criterion methods for assessing body composition.

Even if the interpretation of data on professional dancers (Table 3) should be discussed with caution due to the limited sample size, some stimulating findings emerged: compared to amateurs, they had lower IRs and higher PhA for the upper limbs, suggesting a possible relationship between workout volume and the electrical characteristics of muscle. In addition, smaller IRs and greater PhA for upper limbs were still observed in amateur athletes compared to the controls (Figure 1).

Athletes and controls were studied in standardized conditions by a single experienced operator, while BIA was performed on both body sides to ensure a more reliable assessment of the electrical characteristics of the body. A large proportion of the pole dancers going to two different gyms participated in the study, while control women were selected among those who were enrolled in a study on university students who did low amounts of physical activity.

Indeed, there are limitations to the study that should be considered. It was a single-center cross-sectional study in which body composition was evaluated by means of a field method. Furthermore, we specifically focused on the assessment of raw BIA variables, such as IR and PhA, that are markers of FFM composition or muscle composition and cannot easily be compared with a proper criterion technique. In addition, there was no information regarding participants’ body composition before starting the training, and it was not possible to carry out a very accurate evaluation of the strengthening or conditioning workouts.

## 5. Conclusions

In conclusion, care must be taken not to overinterpret the results of the present study. The main findings were that raw BIA variables that may be considered as markers of FFM composition or muscle composition significantly differed between female pole dancers and controls, suggesting higher BCM, as well as a lower ECW/TBW ratio; in addition, practicing pole dancing is associated with lower FM and BF%.

Differences in PhA and IRs may suggest modifications in the electrical characteristics of the body that seem to be more marked for the upper limbs and possibly in professional than amateur athletes and that was similar for the three IRs considered. These findings are in line with the literature describing changes in raw BIA variables and body composition due to regular physical exercise [8,9,21,22]. Further studies, especially intervention studies, are needed to define the best approach to use BIA in order to measure raw BIA variables and possibly track changes in the body composition of athletes with time.

## Figures and Tables

**Figure 1 ijerph-18-12638-f001:**
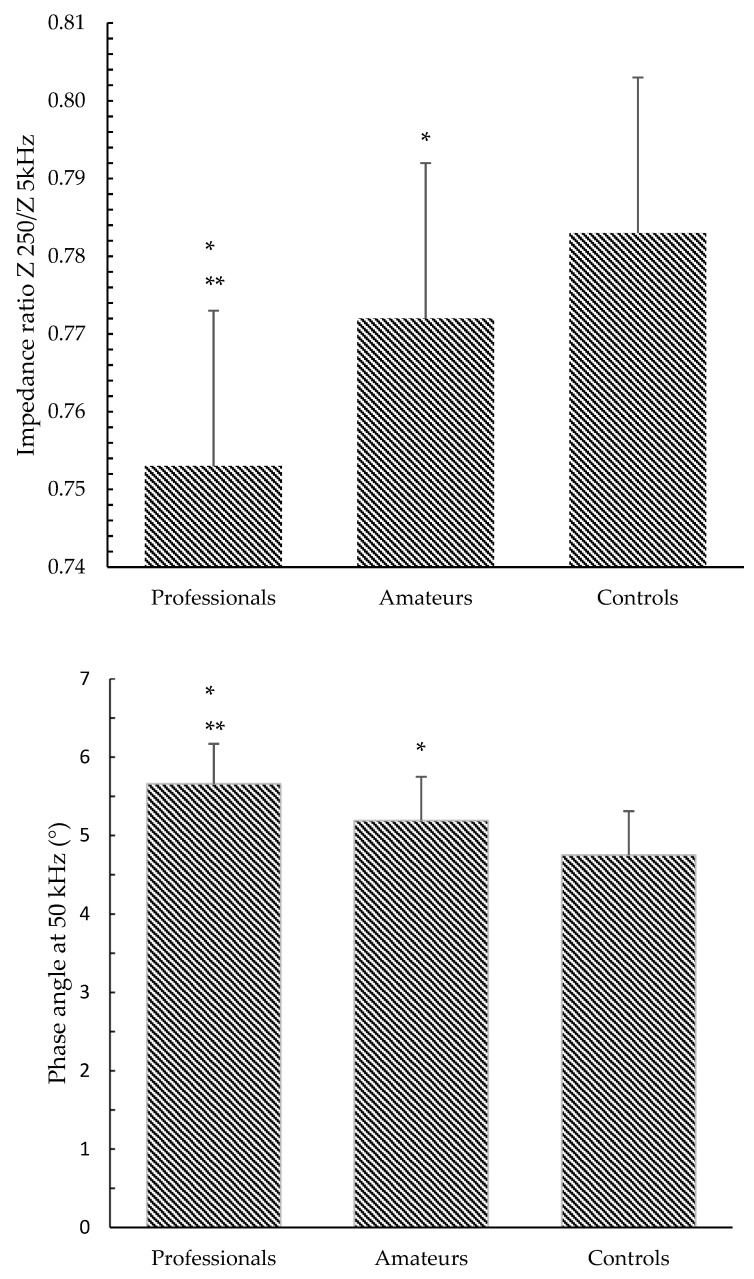
Impedance ratio Z 250 kHz/Z 5 kHz and phase angle at 50 kHz in amateur or professional pole dancers compared to control women. * *p* < 0.05 vs. controls ** *p* < 0.05 vs. amateurs and controls.

**Table 1 ijerph-18-12638-t001:** Individual characteristics and body composition in female pole dancers and controls.

	Pole Dancers (*n* = 40)	Controls (*n* = 59)	*p*-Value	Cohen’s d
Age (years)	27.4 ± 5.1	26.8 ± 4.7	0.561	0.12
Body mass (kg)	57.0 ± 6.9	58.6 ± 6.4	0.225	0.24
Height (cm)	160.3 ± 5.1	161.9 ± 4.9	0.139	0.32
BMI (kg/m^2^)	22.2 ± 2.3	22.3 ± 1.8	0.747	0.05
Fat-free mass, FFM (kg)	43.5 ± 3.5	43.0 ± 3.1	0.448	0.15
Fat-free mass index, FFMI (kg/m^2^)	16.9 ± 1.1	16.4 ± 0.8	0.007 *	0.52
Fat mass, FM (kg)	13.5 ± 4.3	15.6 ± 4.1	0.013 *	0.50
Percentage body fat, BF%	23.2 ± 4.7	26.3 ± 4.4	0.001 *	0.74
Arm muscle area D, AMA (cm^2^)	52.5 ± 9.4	48.9 ± 8.9	0.060	0.39
Arm fat area D, AFA (cm^2^)	2.0 ± 0.5	2.2 ± 0.8	0.047 *	0.30
Arm muscle area ND, AMA (cm^2^)	51.8 ± 10.4	48.0 ± 8.4	0.045 *	0.40
Arm fat area ND, AMA (cm^2^)	2.0 ± 0.6	2.2 ± 0.7	0.098	0.31

Data are expressed as mean ± standard deviation. * *p* < 0.05. BMI—body mass index. FFM and FM were estimated from the BIA; AMA was corrected for bone area. D—dominant side and ND—non-dominant side of the body. Effect size: Cohen’s d ≤ 0.2 = small, 0.2 < d ≤ 0.5 = small to medium, 0.5 < d ≤ 0.8 = medium to large, d > 0.8 = large.

**Table 2 ijerph-18-12638-t002:** Bioimpedance indexes, impedance ratios and phase angles that were measured for the whole body and upper and lower limbs in female pole dancers and controls.

		Pole Dancers (*n* = 40)	Controls (*n* = 59)	*p*-Value	Cohen’s d
Bioimpedance Index (Ω)					
Whole body	5 kHz	41.1 ± 4.2	39.4 ± 3.8	0.043 *	0.42
	50 kHz	46.8 ± 4.9	44.6 ± 4.1	0.018 *	0.49
	100 kHz	49.7 ± 5.3	47.2 ± 4.3	0.013 *	0.52
	250 kHz	53.5 ± 5.7	50.8 ± 4.6	0.011 *	0.52
Impedance Ratio					
Whole body	Z 50/Z 5 kHz	0.878 ± 0.014	0.883 ± 0.014	0.060	0.36
	Z 100/Z 5 kHz	0.827 ± 0.017	0.835 ± 0.017	0.039 *	0.47
	Z 250/Z 5 kHz	0.768 ± 0.018	0.775 ± 0.018	0.058	0.39
Upper limbs	Z 50/Z 5 kHz	0.887 ± 0.013	0.897 ± 0.015	<0.001 *	0.71
	Z 100/Z 5 kHz	0.837 ± 0.016	0.852 ± 0.018	<0.001 *	0.88
	Z 250/Z 5 kHz	0.769 ± 0.019	0.783 ± 0.020	<0.001 *	0.72
Lower limbs	Z 50/Z 5 kHz	0.867 ± 0.018	0.865 ± 0.018	0.451	0.13
	Z 100/Z 5 kHz	0.816 ± 0.022	0.814 ± 0.021	0.718	0.09
	Z 250/Z 5 kHz	0.771 ± 0.025	0.769 ± 0.024	0.765	0.08
Phase Angle (°)					
Whole body		6.07 ± 0.56	5.85 ± 0.56	0.063	0.39
Upper limbs		5.27 ± 0.59	4.76 ± 0.56	<0.001 *	0.89
Lower limbs		7.05 ± 0.70	7.06 ± 0.69	0.974	0.01

Data are expressed as mean ± standard deviation. * *p* < 0.05. BI index—bioimpedance index (height^2^/Z), IR—impedance ratio, PhA—phase angle. Cohen’s d ≤ 0.2—small, 0.2 < d ≤ 0.5—small to medium, 0.5 < d ≤ 0.8—medium to large, d > 0.8—large.

**Table 3 ijerph-18-12638-t003:** Bioimpedance index, impedance ratio and phase angle measured for the whole body and upper and lower limbs in amateur and professional pole dancers.

		Professional Pole Dancers (*n* = 7)	Amateur Pole Dancers (*n* = 33)	*p*-Value	Cohen’s d
Impedance Ratio					
Whole body	Z 50/Z 5 kHz	0.869 ± 0.015	0.879 ± 0.014	0.079	0.70
	Z 100/Z 5 kHz	0.817 ± 0.018	0.830 ± 0.016	0.072	0.76
	Z 250/Z 5 kHz	0.756 ± 0.021	0.771 ± 0.018	0.058	0.77
Upper limbs	Z 50/Z 5 kHz	0.875 ± 0.010	0.889 ± 0.013	<0.001 *	1.2
	Z 100/Z 5 kHz	0.824 ± 0.014	0.840 ± 0.015	<0.001 *	1.1
	Z 250/Z 5 kHz	0.753 ± 0.018	0.772 ± 0.018	<0.001 *	1.1
Lower limbs	Z 50/Z 5 kHz	0.863 ± 0.021	0.868 ± 0.018	0.463	0.26
	Z 100/Z 5 kHz	0.810 ± 0.025	0.817 ± 0.021	0.435	0.30
	Z 250/Z 5 kHz	0.763 ± 0.030	0.772 ± 0.025	0.355	0.33
Phase Angle (°)					
Whole body		6.37 ± 0.57	6.00 ± 0.55	0.117	0.66
Upper limbs		5.66 ± 0.56	5.19 ± 0.56	0.041 *	0.99
Lower limbs		7.11 ± 0.80	7.04 ± 0.70	0.821	0.09

Data are expressed as mean ± standard deviation. BI index—bioimpedance index calculated as height^2^/Z. * *p* < 0.05. Cohen’s d ≤ 0.2—small, 0.2 < d ≤ 0.5—small to medium, 0.5 < d ≤ 0.8—medium to large, d > 0.8—large.

## Data Availability

The datasets used and/or analyzed during the current study are available from the corresponding author on reasonable request.

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
