# Peer review of "Body Composition and Bioelectrical-Impedance-Analysis-Derived Raw Variables in Pole Dancers"

_ijerph, 2021, doi:10.3390/ijerph182312638_

Round 1

Reviewer 1 Report

The article is interesting and fits well within the increased literature on body composition and phase angle in athletes. However, there are some things that could be improved, in my opinion.

Introduction. The introduction is a bit confused, it could be improved following a logical course (first explanation of body composition and then enter into the specific pole dance).

Methods. In line 95, I think te phrase is not so clear. So it seems as if four different sessions were done for each women. In line 116, COPD is an abbreviation, but but there is no writing of what and it cannot be taken for granted that readers know the acronym. 

There are specific guidalines to the degree of arm abducted (line 116-118). 

This is just my consideration, but does it make sense to consider all three frequencies? What does it add to me compared to considering just one frequency?

It’s been pushed to the limit, but having no statistical strength, I think it does not make sense to considerer professionals and amateur separately. It would be better to consider them all together.

Discussion. The discussion, like the introduction, should be reviewed. Not for problems related to the content, but on the logical sequence of the presented data. In addition, some sentences are repeated several times (for example, lines 217-219, 221-224). 

Reviewer 2 Report

The manuscript describes an interesting small-scale study. Presentation is generally acceptable but there a number of points that require attention.

Line 63. The authors refer to phase angle but at what frequency? Other than zero and infinite a phase angle exists at every frequency.

Line 64. Please state what is meant by the vague term "muscle quality". Why should a ratio of impedance vectors relate to "quality".  Impedance (Z) is dominated by resistance (R) and this relates to water. I think of "quality" referring to structure. This would be more indexed by reactance or membrane capacitance.

Line 115. This is not the generally accepted ratio and the citing references do not support this measurement. Impedance ratio is that of Z at frequencies of 5 and 200 not 250 as stated in the citations. This point should be made clear and the use of this ratio justified (not simply that this was the available data from the device). Related to this, It would have been possible to estimate with could precision Z at 200 kHz. Indeed, there are papers that show how such data as provided by the IM Touch could be used to emulate a BIS device, 40-50 frequencies are NOT mandatory as suggested at line 251. 

Line 126. Ref 15 does not actually provide information as to the relative value (or otherwise) of ratios at different frequencies. In fact, many would argue that at 250 kHz. the measured impedance is still dominated by ECW and publications attest to this. Indeed, theory shows that TBW is only related to resistance at infinite frequency and that 50 kHz widely used to predict TBW does so because it is close to the characteristic frequency at which TBW can be predicted for Z. In short, While associations between body composition characteristics and ratios may be observed, these are empirical observations not well grounded in underlying theory. This should be discussed.

Table 2 and results in general. Significant differences between pole dancers and controls were found, e.g., 50 kHz 46.8 ohm versus 44.6 ohm P = 0.018. This is a tiny difference, 2.2 ohm or around 5% Published data suggests that differences of similar magnitude. can be seen in day-to-day variation and in  test-retest studies. Indeed these differences are smaller than the differences between the 40 pole dancers in Table 2 and the group of 7 in Table 3.  In addition what are the effect sizes - e.g., Cohen's d. For the above I calculate this to be 0.48. This is only a small to moderate effect size.  I, therefore question the overall value of the observations. 

Discussion. 

The authors discuss their data with reference to muscle quality (point above) or suggest that their data relate to muscle structure. Where is the support for this? Impedance measurements of the whole body relate to ALL tissues that are conductive not just muscle. For the limbs, impedance may relate more closely just to muscle but structure? Structure relates to the elements of muscle the myofibrillar and cellular form. It would be perfectly possible for a given conductive volume with the same amount of membraneous material but arranged in different ways to provide similar if not identical impedance data when measured simply at a few frequencies with PhA at one frequency only. This could be readily shown from modelling with such as BioZsim (https://upcommons.upc.edu/bitstream/handle/2117/93645/05Aic05de11.pdf?sequence=5&isAllowed=y) Care must be taken not to over interpret the results.

Reviewer 3 Report

The aim of this study was to evaluate the body composition of pole dancers at different levels compared to controls. Authors also studied raw BIA variables for assessing muscle quality.

There are several shortcomings in this study that prevent it from being published in this form. 

  1. Significant differences between the pole dancer group and the control group are highlighted in Table 1. The percentage differences in FM and FFM are evident, and this could explain a large part of the results of the study.
  2. The BIA method has an error of 4-8%. In athletes this risk of error is even more evident. https://www.nature.com/articles/ejcn2012165 The authors should point out that the method used is not ideal for assessing the body composition of athletes.
  3. A large variability in PhA is observed for the same sport, while it is still uncertain to what extent PhA differs between various sports. There are no clear relationships of PhA with sport performance or training/untraining. https://jissn.biomedcentral.com/articles/10.1186/s12970-019-0319-2
  4. The use of raw bioelectrical impedance analysis variables for the assessment of muscle quality has no scientific basis. However, it is evident that the differences between the two groups shown in Table 1 can explain these raw BIA.

Round 2

Reviewer 2 Report

The authors have satisfactorily addressed the issues raised in my original review.

Author Response

We would like to thank the Reviewer for his/her comments and suggestions. We are happy to know that our revisions were satisfactory. 

Reviewer 3 Report

The authors have made some minor changes to the text but in order to be published the work needs to be completely revised in the light of the comments made in my previous review.
